# Challenges to the Fight against Rabies—The Landscape of Policy and Prevention Strategies in Africa

**DOI:** 10.3390/ijerph18041736

**Published:** 2021-02-10

**Authors:** Andrea Haekyung Haselbeck, Sylvie Rietmann, Birkneh Tilahun Tadesse, Kerstin Kling, Maria Elena Kaschubat-Dieudonné, Florian Marks, Wibke Wetzker, Christa Thöne-Reineke

**Affiliations:** 1International Vaccine Institute, Seoul 08826, Korea; sylvie_rietmann@hotmail.com (S.R.); birkneh.tadesse@ivi.int (B.T.T.); fmarks@IVI.INT (F.M.); 2Immunization Unit, Robert Koch Institute, 13353 Berlin, Germany; KlingK@rki.de; 3Institute of Animal Welfare, Animal Behavior and Laboratory Animal Science, Center for Veterinary Public Health, Freie University Berlin, 14163 Berlin, Germany; tierarztpraxis.kaschubat@outlook.de (M.E.K.-D.); Christa.Thoene-Reineke@fu-berlin.de (C.T.-R.); 4Laboratory of Microbiology and Parasitology, University of Antananarivo, 566 Antananarivo, Madagascar; 5Cambridge Institute of Therapeutic Immunology and Infectious Disease, University of Cambridge School of Clinical Medicine, Cambridge Biomedical Campus, Cambridge CB2 0SL, UK; 6Clinic for Anaesthesiology and Intensive Care Medicine, Jena University Hospital, 07743 Jena, Germany; wibke.wetzker@med.uni-jena.de

**Keywords:** rabies, zoonoses, prevention, Africa

## Abstract

Nearly 59,000 human deaths worldwide are attributable to rabies annually, of which more than a third occur in Africa. In recent years, progress has been made in both action and collaboration including implementation of surveillance and prevention measures. In this review we assess the scale of surveillance, preventive, and control efforts of canine-transmitted human rabies in African countries. We reviewed literature published from 2014 to 2018, retrieved from electronic databases including MEDLINE, Global Index Medicus, BIOSIS, Science Citation Index, and EMBASE. WHO reports, national disease control program reports, and conference proceedings were also reviewed. The database search was conducted using keywords including rabies, control, and prevention. In forty countries (40/54), some level of rabies control and prevention strategy was available while in fourteen (14/54) countries, no specific national control and prevention strategy for human rabies could be retrieved. Thirty-four (34/54) countries utilized the Stepwise Approach towards Rabies Elimination (SARE) tool to monitor the national rabies control efforts—five of these countries were at the lowest tier (0/5) of the SARE scoring system while no country had achieved the highest score (5/5). High burden countries need to step up the implementation of context specific national rabies control, prevention, and monitoring strategies. As a zoonosis, rabies control and elimination require coordination between human and veterinarian health sectors under the “One Health” umbrella and with national master plans on the prevention and control of neglected tropical diseases ending in 2020, the time to act is now.

## 1. Introduction

Rabies, which is caused by viruses of the rhabdovirus family, genus lyssaviruses, has a worldwide distribution. Rabies is vaccine preventable; however, there is no cure, and it is usually fatal once symptoms appear [1]. The neurotropic virus is mainly transmitted to humans by dog bites or scratches, causing encephalomyelitis that leads to nearly 59,000 human deaths annually [2]. Nonetheless, the mortality and morbidity associated with rabies is potentially underestimated due to inadequate and/or poorly designed surveillance systems [2,3,4]. Data presented in the World Health Organization (WHO) expert consultative meeting in 2018 reported that 21,476 human deaths occur each year due to dog-mediated rabies in Africa [3]. More importantly, the majority of deaths on the continent occurred in rural areas and in populations living in healthcare underserved regions with mortality in young children being particularly high [2,3,4]. Data from a global economic study estimated the highest cost of human mortality (45%) and lowest investment on post-exposure prophylaxis (PEP) (3.28% of the global non-human mortality cost) in Africa, indicating a high benefit of rabies elimination campaigns, or improved access to PEP [5].

The Sustainable Development Goals (SDGs) of the 2030 agenda for sustainable development called for actions targeting improvements in health and education and ending epidemics of neglected tropical diseases (NTDs) in the next decade [6]. As a deadly NTD and a disease causing unimaginable human suffering and death, rabies prevention, control, and elimination strategies constitute one of the key targets for many countries in Africa. However, despite the national and global interest to eliminate canine rabies, the lack of reliable epidemiologic data hinders a prudent measure of the progress made on the continent.

Investigations from the past five years estimated the epidemiological and economic impact of intervention strategies strongly supporting the effectiveness of canine vaccination and equitable access to PEP in populations residing in endemic regions [7,8,9,10,11]. Understanding the progress of individual countries and the disparity of infrastructure and programs within countries across the continent will be crucial for the implementation of targeted control and prevention strategies for canine rabies. This will further regional and, ultimately, worldwide efforts to eliminate rabies. Government agencies in countries and regions across Africa are, however, facing the challenge of implementing a well-coordinated human-animal-environment interface designed to support effective, local multifaceted programs [12]. Coordination and collaboration regarding rabies control and elimination activities, both within and across countries, are lacking in Africa. Currently, there is little public information concerning policies and strategies addressing rabies elimination for the whole continent.

In this review, we assess the current control strategies, policies, programs, and actions directed at reducing canine-transmitted rabies in Africa. This summarized evidence will inform regional stakeholders, governments, and local officials to coordinate control, prevention, and elimination strategies across relevant governmental and non-governmental sectors to ensure enhanced collaboration in achieving the SDG targets. Further, the review will provide critical insight into the challenges and deficiencies related to current national policies and strategies. This will support development of informed and coordinated national and continental strategies for rabies elimination.

## 2. Materials and Methods

A scoping review of the control and prevention of canine-transmitted rabies in Africa was conducted by searching peer-reviewed articles from MEDLINE, Global Index Medicus, BIOSIS, Science Citation Index, and EMBASE published between 1 January 2014 and 31 December 2018. Search terms related to the disease (rabies, canine rabies), control and prevention efforts (policies, regulations, national action plans, prevention, and control), and the region (Africa, specific country names) were utilized. The selection of African countries was done in reference to the 54 sovereign states which are listed as member states of the United Nations (https://www.un.org/en/member-states/index.html) (accessed on 21 January 2021). Searches were run combining the disease with each of the control and prevention related key words using Boolean words (AND/OR). Reference lists were scanned for relevant review articles and conference proceedings. English and French articles were included in the review. 

World Health Organization (WHO) and national disease control program reports as well conference proceedings were also reviewed. Country specific data on SARE stages were retrieved through the Pan-Africa Rabies Control Network (PARACON) and Middle East and Eastern Europe Rabies Expert Bureau (MEEREB) interfaces of the Global Alliance for Rabies Control (GARC) [13].

We also searched, retrieved, and reviewed specific rabies and generally neglected tropical diseases (NTD) national prevention and control strategies and plans through websites of Ministries of Health of each of the countries. Requests to key stakeholders for unpublished strategies, plans, and reports were also made for countries with no publicly available records. For countries with official languages other than English and French, we included literature, policy documents, and national guidelines translated into English or French.

Quantitative and qualitative data were extracted from all available records in a structured data extraction tool in Microsoft Excel file. Key variables included the availability of a NTD and/or rabies national plan, whether rabies prevention and control is addressed in national plans/strategies, availability of a Stepwise Approach towards Rabies Elimination (SARE) score and human and/or animal vaccination programs. Furthermore, comprehensiveness of national strategic plans and challenges to rabies control and prevention were qualitatively evaluated against specific parameters including specific action items and monitoring plans in the extraction tool (Table 1).

### 2.1. Global Efforts for Rabies Prevention, Control, and Elimination

Under the umbrella of “One Health”, a collaborative, multisectoral, and transdisciplinary approach including human and animal health and environmental scientists, countries are working towards elimination of canine rabies with the support of the WHO, the World Organization for Animal Health (OIE), the Food and Agriculture Organization of the United Nations (FAO), and GARC [15]. This effort shares the same goals as the Sustainable Development Goals (SDG) aiming to end epidemics of neglected tropical diseases by 2030, as stated by the United Nations in 2015 [6]. FAO, OIE, WHO, and GARC collaborated in 2012 to develop a tool to increase rabies control through (I) effective use of vaccines in animals, medicines, tools, and technologies; (II) generating, innovating, and measuring impact; and (III) sustaining commitment and resources [16]. All partner organizations of the “United Against Rabies Collaboration” were involved in the conception and support the implementation of the resulting SARE tool, through training in more than 40 countries, 36 of which are on the African continent (https://rabiesalliance.org/tools/planning-tools/sare) (accessed on 21 January 2021). The SARE tool guides countries in the formal evaluation of their current status of rabies control [17]. Operational guidance is provided by the GARC through the “*Blueprint for Rabies Prevention and Control*” [18]. Together, the blueprint and the SARE tool enable countries to systematically build up infrastructures and multisectoral collaborations while measuring their progress towards a rabies free country status. 

PARACON, an advisory and networking initiative, was formed under the secretariat of GARC to unify all sub-Saharan African countries towards a One Health approach for rabies control and elimination [13]. This regional rabies surveillance platform acts as a centralized data collection and analysis tool, where members of PARACON submit their data, with optional sharing of information publicly (https://rabiesalliance.org/networks/paracon) (accessed on 21 January 2021). Another notable effort under the GARC is the Middle East, Eastern Europe, Central Asia, and North Africa Rabies Control Network (MERACON), that arose out of the Middle East and Eastern Europe Rabies Expert Bureau (MEEREB) in 2015, which unites scientific expertise and national stakeholders to inform prevention programs and complements the coverage of the African continent. 

### 2.2. Vaccination Programs in African Countries

The level of rabies awareness and adapted strategies for surveillance and prevention varied widely across countries. Thirty-nine (39/54, 70.4%) African countries have implemented an official vaccination program for animal vaccination against rabies. Vaccination programs in 15 of these countries focus on dog vaccination while in 10 countries, both dog and cat vaccinations are practiced. Figure 1 provides an overview of the distribution of vaccination programs targeting cats, dogs, domestic animals (i.e., animals that live in contact with and under control of or owned by humans including buffalo, cattle, goats, pigs, rabbits, sheep), and wildlife (i.e., animals that grow and live in the wild) in African countries.

### 2.3. National Rabies Prevention and Control Plans

National efforts for rabies control and prevention need to be coordinated through strategic plans, which outline specific, attainable, and measurable actions; monitoring strategies; and the financial provisions for all the planned activities. Five countries including Algeria, Angola, Ivory Coast, Kenya, and Namibia have an accessible and specific national strategic plan for the elimination of rabies (Table 1). In Algeria, rabies control programs dating back to 1996 have been described and include vaccination campaign of cats, dogs, and cattle incorporating interdisciplinary partnerships under the One Health approach. The country recently proposed to improve the impact of interventions [19,20].

Fourteen countries (14/54, 25.9%) included rabies prevention and control measures in their NTD Masterplan (Table 1). Eritrea [21], Ghana [22], Liberia [23], Mali [24], Niger [25], Nigeria [26], and South Sudan [27] were targeting the development or enhancement of surveillance systems and improvement of infrastructures, addressing the present data gaps. 

The Central African Republic [28], DR Congo [29], the Gambia [30], Guinea Bissau [31], Senegal [32], Kenya [33,34], Mali [24] and Swaziland [35] specified activities such as rabies case management, capacity building, collaboration between different ministries or national reference laboratories, surveillance, laboratory surveillance, PEP, or vaccination of domestic animals. Senegal compiled a comprehensive budget plan for rabies prevention and control measures as well as other NTD [32]. Similarly, the national plan of Kenya provides a detailed budget plan for elimination measures [34]. Whereas, in Mali, a budget was indicated without specifying the activities [24].

The national program against rabies of the Kingdom of Morocco targets reduction of rabies incidence by 50% in 2021 and elimination of canine-transmitted rabies by 2025 [36]. In Tunisia, the national strategy against rabies was started in 1981, including canine mass vaccination in the form of regular campaigns and as outbreak response, laboratory confirmation of veterinary and human cases, and public education programs [37].

Tanzania initiated a multi-faceted project in 2010, which was supported by the WHO in 28 districts on the Tanzanian mainland and the island of Pemba and in 2017 gave impetus to the development of a National Rabies Control and Elimination Strategy aiming at control of dog rabies incidence and elimination of human rabies by 2030 [38,39]. Similarly, in Rwanda, rabies control is attempted through annual campaigns for vaccination of owned dogs in addition to culling stray dogs [40,41]. The NTD Master Plan of Uganda was not accessible; therefore, their roadmap for rabies prevention and control strategy is unknown. However, Uganda has a One Health Strategic Plan (2018–2022), which includes rabies as one of the seven priority zoonoses [42]. 

Botswana and Ethiopia have no national rabies plan nor integrated measures in their respective NTD master plan and for five countries including Benin, Burkina Faso, Burundi, Cabo Verde, and Cameroon, no data was publicly available (Table 1).

In four countries, namely, Angola, Ivory Coast, Namibia, and Sierra Leone, national plans are mentioned in peer-reviewed publications or from sources linked to national policies or by the scientific community [43,44,45,46].

### 2.4. Challenges Identified in National Plans 

Identification of the key challenges to implementation of rabies prevention and control strategies is an essential first step to successfully achieving regional, national, and global targets. Although most of the African countries are members of one of the rabies networks such as MEEREB or PARACON, national rabies plan, stand-alone or as part of the NTD master plan are not yet implemented. Qualitative assessment of existing plans is complicated by the diversity of country-specific definitions of prevention or control measures such as monitoring and surveillance or limited simply due to lack of data (Table 1). Furthermore, attention to implementation challenges countries have experienced with rabies control efforts have been given little consideration or are not often reported. Equatorial Guinea’s strategy focused on the conduct of a national rabies prevalence survey and did not include other prevention measures [47]. Malawi and the Seychelles indicated rabies in the national NTD Master Plan as one of the 17 NTD identified by the WHO, but their respective plans merely address rabies as a public health threat [48,49].

The lack of epidemiologic data is a significant challenge in streamlining rabies control and prevention. In Mali and Nigeria, despite having more elaborate strategies within the NTD Master Plan, the scarcity of prevalence and incidence data on rabies poses significant challenges to the control and prevention efforts [24,26]. In contrast, epidemiological surveillance programs conducted by the national veterinary services in collaboration with the human health sector contributed to enhanced prevention and control efforts in Niger [25]. The NTD master plan, however, points out the lethargy in further cross-sectoral implementation and the absence of the coordination of activities [25]. Whereas in the Kingdom of Lesotho, which sponsors annual dog vaccination campaigns, the control was reported to be complicated by estimated low vaccination coverage, lack of epidemiological understanding and limited surveillance data [50].

In Ethiopia, despite not publishing a national rabies elimination plan, a surveillance network exists including the Public Health Emergency Management of the Ethiopian Public Health Institute, the Ministry of Livestock and Fisheries, and Ethiopian Wildlife Conservation Authority [51].

### 2.5. Monitoring Progress in Rabies Prevention, Control, and Elimination 

Monitoring progress of national programs is at the heart of the elimination efforts. Thirty-five African countries, as well as the semi-autonomous region Zanzibar, went through the self-assessment process with SARE and shared their score with PARACON including open access of data (see Figure 2). Generally, countries first report at Stage 0 of the SARE scoring system when little or no epidemiological data exists and control efforts for rabies are lacking or scarce [17,18]. Countries would be expected to progress throughout the next stages, until eventually reaching Stage 5 defined as freedom from human and dog-transmitted rabies being monitored [52]. Key activities include vaccination of canine populations, rabies awareness and communication campaigns, access to PEP, and capacity building for standardized rabies surveillance.

### 2.6. Future Perspectives in Rabies Prevention, Control, and Elimination

The current level of implemented programs and activities shows that rabies is still a neglected disease and surveillance, prevention and control measures are still inadequate on the African continent. It is important for national and international stakeholders to understand that rabies has the highest case fatality ratio of all infectious human diseases and, more importantly, that human death related to canine-mediated rabies could be 100% prevented by enhanced vaccine coverage in the canine population. 

For example, in South Africa, current surveillance for rabies was described as meagre with inadequate dog rabies vaccine coverage in many areas [53]. The decrease in human rabies cases in KwaZulu-Natal related to the efforts in dog rabies control in this province from 2009 to 2015 highlighted the link between dog rabies and the occurrence of human cases. The collapse in control efforts in this province, however, led to the incidence of 2, 7, and 8 human cases in 2016, 2017, and 2018, respectively [53]. 

Our findings agree with those of other reviews that identified shortages in capacity to prevent and control canine-mediated rabies in Africa [51]. The relatively little attention given to a coordinated control and prevention of rabies makes the global target of achieving elimination by 2030 challenging. Now is the time to act as the majority of NTD master plans or national plans are ending in 2020. Recognizing the complexity of the disease and transmission mode would better inform prevention and control methods. The design of effective rabies elimination strategies and implementation should span healthcare sectors including veterinarians, public health workers, physicians, ecologists, and vaccine producers. Synchronization of activities through a multi-disciplinary approach will increase the chances of overcoming technical and infrastructural barriers towards regional forces that drive disease control under the One Health approach. 

Canine-mediated rabies should not be recognized as a neglected tropical disease for which endemicity is regionally limited to the African continent but instead be understood as a global infectious disease problem calling for more action and coordination worldwide. As with many infectious diseases, rabies does not respect national borders and can impact any country in the world given the unprecedent air travel. A recent review investigated cases of rabies in international travelers, expatriates, and migrants and found two-thirds of identified cases in migrants from rabies-endemic low-and middle-income countries [52]. As long as endemic countries exist, rabies will continue to be a far-reaching, global health concern. The success in global eradication of smallpox, which was achieved through strong global efforts and coordination of strategies, logistics, and vaccine investments is a very good role model for many other infectious diseases including rabies [53]. Similarly, the goal for global elimination of rabies by 2030 demands that the public health community continues its momentum and governance mechanisms towards this effort.

## 3. Limitations

The data included in this review was retrieved from peer-reviewed journals and publicly accessible literature. We do not claim full coverage of all available information on every African country since the search was limited to data available in English and French even though most governmental portals provide English versions. Country specific regulations, practices, national strategies, or plans might not have been shared publicly or might not have been accessible and hence were not included in this review.

## Figures and Tables

**Figure 1 ijerph-18-01736-f001:**
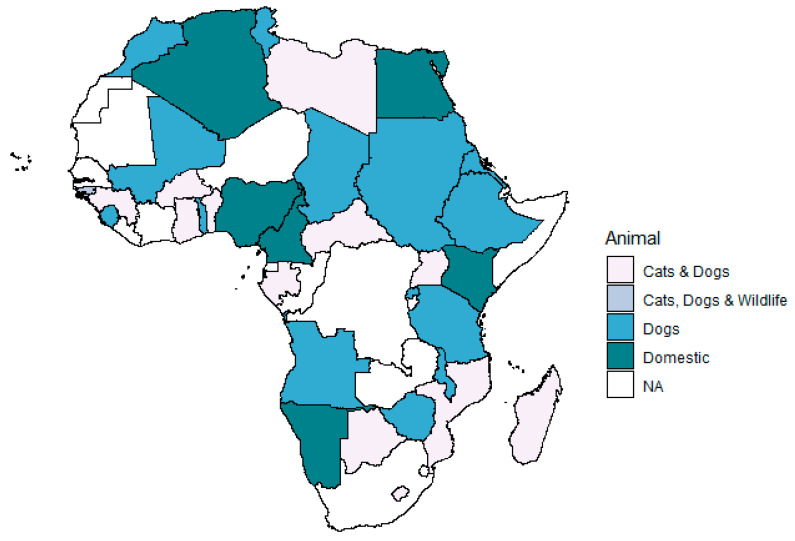
Official rabies vaccination program in Africa by 2019. NA: not available, represents countries with no official plan or no data retrieved. Domestic refers to animals that live in contact with and under control of or owned by humans and summarizes here species including buffalo, cattle, goats, pigs, rabbits, and sheep; see details per country in Table 1; Wildlife refers to animals that grow and live in the wild [18].

**Figure 2 ijerph-18-01736-f002:**
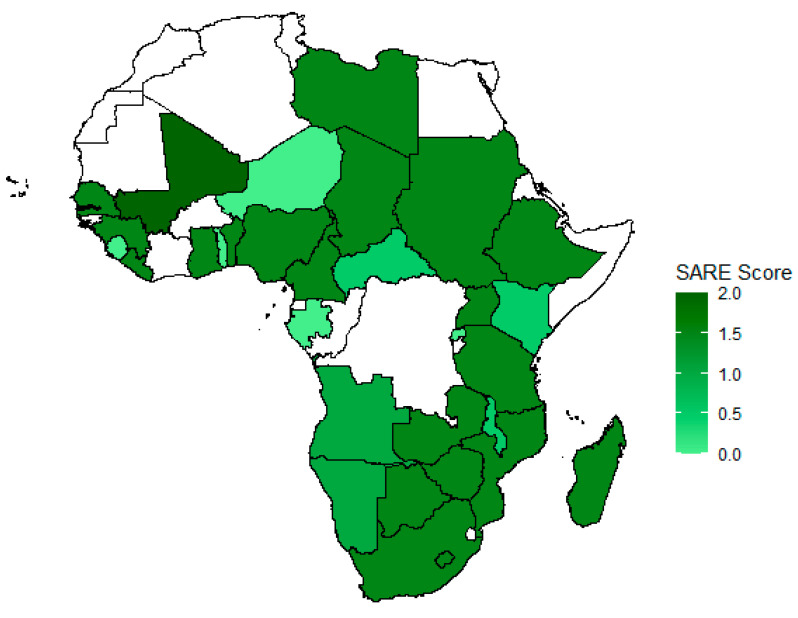
Distribution of SARE scoring levels across Africa as of 2019. Note: Countries where no data was available are marked white.

**Table 1 ijerph-18-01736-t001:** Summary of extracted key variables for 54 African countries.

Country	Part of Rabies Network	National Rabies Plan (Yes/No)	NTD Master Plan (Yes/No, Time Period)	Rabies Addressed in NTD Master Plan (Yes/No)	Disease Notification (Domestic/Wild/Species Specified)	Precaution at Borders (Domestic/Wild/Species Specified)	Monitoring	Screening	General Surveillance	Targeted Surveillance	Official Vaccination	SARE Score
Algeria	MEEREB	yes	†	†	D	W/D	D	†	†	†	cattle, dogs, cats	†
Angola	PARACON	yes	†	†	W/D	D	W/D	†	W/D	D	dogs	1
Benin	PARACON	†	†	†	W/D	D	D	†	D	†	cats, dogs	1.5
Botswana	PARACON	†	Yes (2015–2020)	no	W/D	dogs, cats, sheep	W, cattle	†	cats, dogs, cattle, goats, sheep/W	W	cats, dogs	1.5
Burkina Faso	PARACON, no data shared	†	†	†	D	cats, dogs	†	†	†	†	cats, dogs	†
Burundi	PARACON, no data shared	†	†	†	†	†	†	cattle, cats, dogs	cattle, cats, dogs	†	†	†
Cabo Verde	†	†	†	†	D	D	†	†	†	†	†	†
Cameroon	PARACON	†	†	†	D	†	†	†	†	dogs	D	1.5
Central African Republic	PARACON	†	†	†	D	D	†	cats, dogs	cats, dogs	†	cats, dogs	0.5
Chad	PARACON	†	†	†	dogs	dogs, cats, sheep	dogs	dogs	†	†	dogs	1.5
Comoros	†	†	†	†	dogs	†	†	†	dogs	†	†	†
Democratic Republic of the Congo	PARACON	†	Yes (2016–2020)	yes	cats, dogs	†	cats, dogs	cats, dogs	cats, dogs	†	D	1.5
Djibouti	PARACON, no permission to share data	†	†	†	cattle, cats, dogs	cattle, cats, dogs	†	†	†	†	†	†
Egypt	†	†	†	†	W/D	W/D	†	†	†	W/D	D	†
Equatorial Guinea	PARACON, no data shared	†	Yes (2018–2022)	yes	D	D	†	†	†	†	†	†
Eritrea	PARACON, no data shared	†	Yes (2015–2020)	yes	dogs, cattle, equidae	†	†	†	dogs, cattle, equidae	†	dogs	†
Eswatini (Swaziland)	†	†	Yes (2015–2020)	yes	W/D	W, cats, dogs	†	†	W/D	†	dogs	†
Ethiopia	†	†	Yes (2016–2020)	no	D	cats, dogs	†	†	D	†	dogs	1.5
Gabon	PARACON, no permission to share data	†	†	†	†	†	†	†	†	†	cats, dogs	0
Gambia	†	†	Yes (2015–2020)	yes	†	†	†	†	†	†	D	†
Ghana	PARACON	†	Yes (2016–2020)	yes	D	†	D	†	D	†	cats, dogs	1.5
Guinea	PARACON, no permission to share data	†	†	†	D	D	†	†	D	†	cats, dogs	1.5
Guinea-Bissau	PARACON	†	Yes (2014–2020)	yes	cats, dogs	cats, dogs	cats, dogs	†	cats, dogs	dogs	cats, dogs, W	†
Ivory Coast	PARACON	yes	Yes (2016–2020)	no	D	D	†	†	D	†	cats, dogs	2
Kenya	PARACON	yes	Yes (2016–2020)	no	D	D	D	†	W/D	†	equidae, cattle, sheep, swine, dogs	0.5
Lesotho	PARACON	†	†	†	D	cats, dogs	†	†	D	†	cats, dogs	1.5
Liberia	PARACON	†	Yes (2016–2020)	yes	†	†	†	†	†	†	†	1.5
Libya	MEEREB, PARACON	†	†	†	D	D	†	†	†	†	cats, dogs	1.5
Madagascar	PARACON	†	†	†	D	D	†	†	D	†	cats, dogs	1.5
Malawi	PARACON	†	Yes (2015–2020)	yes	W/D	†	†	dogs	W/D	†	dogs	0.5
Mali	PARACON	†	Yes (2017–2021)	yes	dogs	dogs	†	†	†	†	dogs	2
Mauritania	PARACON, no data shared	†	†	†	D	†	D	†	D	†	†	†
Mauritius		†	†	†	†	†	†	†	†	†	†	†
Morocco	MEEREB	yes	yes (†)	†	D	cats, dogs	D	†	D	D	dogs	†
Mozambique	PARACON	†	†	†	D	D	D	†	D	D	cats, dogs	1.5
Namibia	PARAVON, no permission to share data	yes	Yes (2015–2020)	no	D	D	D	†	D	†	cattle, dogs, cats	1
Niger	PARACON	†	Yes (2016–2020)	yes	D	†	†	†	D	†	†	0
Nigeria	PARACON	†	Yes (2015–2020)	yes	cattle, cats, dogs	†	†	†	cattle, cats, dogs	†	buffalos (D)	1.5
Republic of the Congo	PARACON no permission to share data	†	†	†	W/D	W/D	W/D	†	W/D	W/D	cats, dogs, equidae	1
Rwanda	PARACON	†	†	†	D	D	D	D	†	D	dogs	0
Sao Tome and Principe	†	†	†	†	†	†	†	†	cattle, dogs, cats, hares, rabbits, sheep, goats, swine	†	†	†
Senegal	PARACON	†	Yes (2016–2020)	yes	cattle, cats, dogs	cattle, cats, dogs	cattle, cats, dogs	†	cattle, cats, dogs	†	†	1.5
Seychelles	†	†	†	†	D	D	†	†	D	†	†	†
Sierra Leone	PARACON	yes (not accessible)	†	†	dogs	†	†	†	dogs	†	dogs	0
Somalia	PARACON, no permission to share data	†	†	†	D	D	D	D	†	†	†	†
South Africa	PARACON	†	†	†	W/D	W/D	†	†	W/D	†	†	1.5
South Sudan	†	†	Yes (2016–2020)	yes	D	D	†	†	†	†	dogs	†
Sudan	PARACON	†	†	†	D	†	†	†	†	†	dogs	1.5
Tanzania	PARACON	†	†	†	D	†	W	†	D	W	dogs	1.5
Togo	PARACON, no permission to share data	†	†	†	dogs	dogs	dogs	dogs	dogs	dogs	dogs	0
Tunisia	MEEREB	†	†	†	W/D	W/D	†	†	W/D	†	dogs	†
Uganda	PARACON	yes (not accessible)	†	†	D	D	†	†	D	†	cats, dogs	1.5
Zambia	PARACON, no permission to share data	†	†	†	W/D	cats, dogs	†	†	W/D	†	†	1.5
Zimbabwe	†	†	†	†	cattle, dogs, goats, cats, swine	cats, dogs	†	†	cattle, goats, cats, swine	†	dogs	1.5

* Note: † indicates no official information was publicly available or no data retrieved; D stands for Domestic refers to animals that live in contact with and under control of or owned by humans and W stands for Wildlife refers to animals that grow and live in the wild [14]. Country was defined as the 54 sovereign states which are member states of the United Nations; Disease Notification was defined as National legal obligation to report any suspected or confirmed case of the disease, infection, or infestation to the relevant authorities; Precautions at borders was defined as measures applied at airports, ports, railway stations or road check-points open to international movement of animal, animal products, and other related commodities, where import inspections are performed to prevent introduction of the disease, infection or infestation into a country/territory or zone; Monitoring was defined as intermittent performance and analysis of routine measurements and observations, aimed at detecting changes in the environment or health status of a population; Screening was defined as survey carried out within the framework of a control program for the disease, infection, or infestation for health qualification of herds/flocks in all or part of the national territory; Surveillance was defined as surveillance not targeted at a specific disease, infection, or infestation, also called passive surveillance. Targeted Surveillance was defined as surveillance targeted at a specific disease, infection or infestation, also called active surveillance. MEEREB, Middle East and Eastern Europe Rabies Expert Bureau; PARACON, Pan-Africa Rabies Control Network; SARE, Stepwise Approach towards Rabies Elimination. The scoring system ranges from 0 (lowest tier) to 5 (highest tier). Stage 0: No information on rabies available, but rabies is suspected to be present. Stage 1: Assessment of the local rabies epidemiology, elaboration of a short-term rabies action plan. Stage 2: Development of a national rabies prevention and control strategy. Stage 3: Full-scale implementation of the national rabies control strategy. Stage 4: Maintenance of human rabies freedom, elimination of dog rabies. Stage 5: Freedom from human and dog-transmitted rabies being monitored.

## Data Availability

No new data were created or analyzed in this study. Data sharing is not applicable to this article.

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
