# Peer review of "Challenges to the Fight against Rabies—The Landscape of Policy and Prevention Strategies in Africa"

_ijerph, 2021, doi:10.3390/ijerph18041736_

Round 1

Reviewer 1 Report

Thank you for your work summarizing the rabies control and elimination strategies and challenges associated with these efforts in Africa.  Most of my edits are focused on the use of English, although I have made some changes hoping that I am assisting with clarity of some concepts.  I do think it would be important to clarify the SARE tool scoring.  There seems to be an inconsistency between the first table and section 2.5 of the manuscript in this regard as my comments will further articulate.  

Thank you for the opportunity to review this work.  

Reviewer 2 Report

The manuscript "Challenges to the Fight against Rabies—The Landscape of Surveillance and Prevention Strategies in Africa" reviews the published peer review literature and other sources in the grey literature. The piece is interesting and timely, but it will be highly improved if recommendations for the elaboration of reviews according to the equator network are considered. It is also imperative authors recognize early on their biases by limiting their search to documents in English and French. More detailed comments go below.

Comments

1) In Line 54 authors indicate that "English and French articles were included in the review." If that is the case the strategy heavily biases the inclusivity of information, considering the many nations whose primary language is arabic, portuguese or even spanish thinking about Equatorial Guinea. If this was the case, it will be best if the authors clarify, starting by the title, that they just focused in the anglophone/francophone Africa. This point also needs to be articulate in lines 100-104. This issue also extends to table 1. The information might be publicly available, but on a different language. Also, some of the information might be available "in print", but not posted on a website. A better methodology would have been to make a call or use the contacts from Paracon and MEEREB. This is again evident lines 180-192, where only Guinea Bissau seems to be a country outside the anglo/francosphere. 

2)In Table one carefully check all entries. For example, Botswana was missing a closing ")" in one of the columns.

3) Line 163, what other domestic animals are considered? Please consider moving here the information from the legend of figure 1.

4) The results seem to not report whatever interesting stuff was in the databases when searching them. A more formal section on the numbers of papers, etc, would be nice to include. Please, refer to the equator network advise on reviews: 

https://www.equator-network.org/reporting-guidelines/prisma-scr/

5) Lines 295-299, see comment 1 above. There might be a strong bias just by filtering things by language and looking at Ministry of Health/Health Agency websites, and assuming information will be posted there.

Author Response

Thank you very much for your comprehensive review. Please see the attachment.

This manuscript is a resubmission of an earlier submission. The following is a list of the peer review reports and author responses from that submission.